# The Role of Co-Deleted Genes in Neurofibromatosis Type 1 Microdeletions: An Evolutive Approach

**DOI:** 10.3390/genes10110839

**Published:** 2019-10-24

**Authors:** Larissa Brussa Reis, Andreia Carina Turchetto-Zolet, Maievi Fonini, Patricia Ashton-Prolla, Clévia Rosset

**Affiliations:** 1Laboratório de Medicina Genômica, Centro de Pesquisa Experimental, Hospital de Clínicas de Porto Alegre, Porto Alegre, Rio Grande do Sul 90035-903, Brazil; brussareis@gmail.com (L.B.R.); maievifonini@hotmail.com (M.F.); pprolla@gmail.com (P.A.-P.); 2Programa de Pós-Graduação em Genética e Biologia Molecular, Departamento de Genética, UFRGS, Porto Alegre, Rio Grande do Sul 91501-970, Brazil; aturchetto@gmail.com; 3Serviço de Genética Médica do Hospital de Clínicas de Porto Alegre (HCPA), Porto Alegre, Rio Grande do Sul 90035-903, Brazil

**Keywords:** genotype–phenotype correlations, microdeletions, neurofibromatosis type 1, phylogenetic analysis

## Abstract

Neurofibromatosis type 1 (NF1) is a cancer predisposition syndrome that results from dominant loss-of-function mutations mainly in the *NF1* gene. Large rearrangements are present in 5–10% of affected patients, generally encompass *NF1* neighboring genes, and are correlated with a more severe NF1 phenotype. Evident genotype–phenotype correlations and the importance of the co-deleted genes are difficult to establish. In our study we employed an evolutionary approach to provide further insights into the understanding of the fundamental function of genes that are co-deleted in subjects with *NF1* microdeletions. Our goal was to access the ortholog and paralog relationship of these genes in primates and verify if purifying or positive selection are acting on these genes. Fourteen genes were analyzed in twelve mammalian species. Of these, four and ten genes showed positive selection and purifying selection, respectively. The protein, RNF135, showed three sites under positive selection at the RING finger domain, which may have been selected to increase efficiency in ubiquitination routes in primates. The phylogenetic analysis suggests distinct evolutionary constraint between the analyzed genes. With these analyses, we hope to help clarify the correlation of the co-deletion of these genes and the more severe phenotype of NF1.

## 1. Introduction

Neurofibromatosis type 1 (NF1) (OMIM # 162200) is an autosomal dominant tumor predisposition syndrome affecting both sexes in all ethnic groups, with an estimated incidence of one per 3000 [1]. NF1 results from dominant loss-of-function (haploinsufficiency) mutations mainly in the *NF1* gene, but this is not the only mutational event that explains the phenotype. NF1 clinical diagnosis is based on criteria approved by the National Institutes of Health Consensus Development Conference [2]. The NF1 phenotype is extremely variable and one possible explanation for this variation is a large number of different mutations in different regions of the *NF1* gene [3]. Nearly 20% of *NF1* mutations are single or multiexon deletions or duplications; 5–10% of these *NF1* deletions are known as microdeletions, surrounding *NF1* and its bordering genes. Four types of microdeletions (type 1, 2, 3, and atypical) have been identified and are different in breakpoint location and size [4]. Patients with microdeletions have been described to have a higher risk of malignant peripheral nerve sheath tumors, lower average intelligence, connective tissue dysplasia, skeletal malformations, dysmorphic facial features, cardiovascular malformations, and a higher burden of cutaneous neurofibromas and previous onset of benign neurofibromas [5,6,7,8,9]. Some authors suggest that increased malignancy may be elucidated by variations in the expression of tumor suppressor genes placed in co-deleted regions [10,11], as some of the genes in the deletion interval may have tumor suppressive functions. The analysis of conserved domains in genes in this cluster indicates that many of these genes are presumably involved in the regulation of cell growth and morphology [12]. NF1 clinical symptoms are variable even when comparing individuals with *NF1* microdeletions, probably due to the variability characteristic of all patients with NF1, regardless of the type of *NF1* gene mutation involved, but also due to changes in deletion size and breakpoint location [13]. Nonetheless, some variability in expression of the clinical symptoms has been detected even within the group of patients that are hemizygous for the same number of genes [14], which could be explained by the expression level of non-deleted genes, as well as by environmental factors. All these findings make the study of genotype–phenotype correlations and the determination of the importance of the co-deleted genes difficult. For this reason, a new approach to investigate the importance of these genes has been proposed by this study, using evolutionary and phylogenetic analyses.

Evolutionary analyses take into account the natural selection, which includes purifying or negative selection and positive selection or neutrality. Most of the genes are affected by purifying selection that decreases the frequency of mutations that disadvantage carriers in a given environment [15,16]. Interspecies neutrality tests use data concerning the divergence between closely related species to detect relatively ancient selective events. Interspecies tests include the non-synonymous and synonymous (dN/dS) test that detects selection acting on protein-coding loci by comparing the ratio of non-synonymous (dN) to synonymous (dS) substitutions [17]. Phylogenetic analyses are important because they enrich our understanding of how genes, genomes, and species evolve. The evolutionary history of living species is usually inferred through the phylogenetic analysis of molecular and morphological information using assorted mathematical models.

In the present study we employed an evolutionary approach to provide further insights into the understanding of the fundamental function of genes that are co-deleted in patients with NF1 microdeletions. Our main goal was to access the ortholog and paralog relationship of these genes in primates and verify if purifying or positive selection are acting on these genes. With these analyses, we hope to help to clarify the correlation of the co-deletion of these genes and the more severe phenotype of NF1.

## 2. Methodology

### 2.1. Background

In a previous study conducted by our group ninety-three unrelated NF1 probands who met the NIH diagnostic criteria were recruited at the Oncogenetics clinics of the Hospital de Clínicas de Porto Alegre, located in the state of Rio Grande do Sul, southern Brazil. The study was approved by the Institutional Ethics Committee Board of Hospital de Clínicas de Porto Alegre, under registration number 13-0260, Clinical evaluation and molecular characterization were enrolled in these patients. We found large gene rearrangements in four patients (6%) by multiplex ligation-dependent probe amplification (MLPA) analysis followed by confirmation using chromosomal microarray [18]. One patient presented the classical 1.4 Mb type 1 microdeletion, spanning from *SUZ12P1* to *LRRC37B*, including 22 genes (16 functional). Two patients showed a similar deletion, but their breakpoints differed slightly (one within *CRLF3* intron 1 and *SUZ12* intron 5, including 19 genes and spanning 1.15 Mb and the other within *CRLF3* intron 4 and the *SUZ12* gene, including 18 genes and spanning 1.13 Mb). Finally, one patient showed an atypical microdeletion, with breakpoints between *SUZ12P1* and *LRRC37BP*, spanning 890 kb and 15 genes. The fourteen genes deleted in at least one of the four patients were selected for the phylogenetic and selection analyses in this study: *CRLF3, ATAD5, TEFM, ADAP2, RNF135, NF1, OMG, EVI2B, EVI2A, RAB11FIP4, COPRS, UTP6, SUZ12*, and *LRRC37B.*

### 2.2. Genes and Organisms Selected for Analysis

The following fourteen genes that are frequently co-deleted in NF1 patients with microdeletions were included in the analyses: cytokine receptor like factor 3 (*CRLF3*), ATPase family AAA domain containing 5 (*ATAD5*), transcription elongation factor, mitochondrial (*TEFM*), ArfGAP with dual PH domains 2 (*ADAP2*), ring finger protein 135 (*RNF135*), neurofibromin 1 (*NF1*), oligodendrocyte myelin glycoprotein (*OMG*), ecotropic viral integration site 2B (*EVI2B*), ecotropic viral integration site 2A (*EVI2A*), RAB11 family interacting protein 4 (*RAB11FIP4*), coordinator of PRMT5 and differentiation stimulator (*COPRS*), UTP6 small subunit processome component (*UTP6*), SUZ12 polycomb repressive complex 2 subunit (*SUZ12*), and leucine rich repeat containing 37B (*LRRC37B*). As the focus of evolutionary analysis was to perform evolutionary analysis of the deleted genes in primates, our ingroup included eight primate species (*Homo sapiens*, *Pan troglodytes*, *Pongo abelii*, *Gorilla gorilla gorilla*, *Nomascus leucogenys*, *Macaca mulatta*, *Callithrix Jacchus*, and *Tarsius syrichta*). The *Canis lupus familiaris*, *Bos taurus*, *Mus musculus*, and *Rattus norvegicus* were used as outgroup. All the included species have their complete genome sequence available in online databases.

### 2.3. Sequence Alignment and Phylogenetic Analyses

In order to obtain gene sequences in the organisms described above, BlastX was performed for each gene from each organism in the Ensembl Databases (EMBL-EBI and Wellcome Trust Sanger Institute) and the National Center for Biotechnology Information (NCBI) using human sequences encoding the 14 genes as queries. The human genes are located in the frequently deleted region of chromosome 17 and were obtained from the Metazome database (Joint Genome Institute and Center for Integrative Genomics) and confirmed in NCBI. Full-length coding sequencing (CDS) and protein sequences were aligned using MUSCLE [19] implemented in MEGA 7.0 [20] with default parameters. The multiple alignments were manually inspected and edited when necessary in the final analysis. The phylogenetic relationships were reconstructed following nucleotide and protein sequence alignments using a Bayesian method carried out in BEAST v1.8.4 [21]. JModelTest 2.1 [22] and ProTest 2.4 [23] were used to select the best model of nucleotides and protein evolution. The birth-death process was selected as a tree prior to Bayesian analysis, and it was run for 10,000,000 generations with Markov chain Monte Carlo (MCMC) algorithms for both amino acid and nucleotide sequences. Tracer v1.6 [24] was used to verify the convergence of the Markov chains and the adequate effective sample sizes (>200) and the normal distribution curve. Trees off the curve were burned by tree annotator with a posterior probability limit of 0.005. Trees were visualized and edited using FigTree v1.4.3 [25].

### 2.4. Selection Analysis

Using the alignments of each gene and the respective phylogenetic trees as inputs, the rates of nonsynonymous to synonymous substitutions (dN/dS or ω), homogeneity, and positive selection could be determined using maximum-likelihood models in the program CodeML in PAML v4.9 [26]. First, we used the models allowing ω to vary among sites (site models). The most effective models were: the basic model (M0), which estimates uniform ω ratio among all sites; the site models including M1 (nearly neutral), M2 (selection), M3 (discrete), M7 (beta distribution, ω > 1 disallowed) and M8 (beta distribution, ω > 1 allowed). The likelihood-ratio test (LRT) test is obtained by calculating twice the log-likelihood difference between the alternative and null models (2ΔL). The LRT was performed between the following pairs of the models: M0 vs. M3; M1 vs. M2, and M7 vs. M8. These LRT statistics approximately follow a chi-square distribution, and the number of degrees of freedom is equal to the number of additional parameters in the more complex model [27]. A significantly higher likelihood of the alternative model compared to the null model suggests positive selection. CodeML was also used to estimate different values of ω among the sites and branches according to the branch site model comparing the alternative model (model = 2, Nsites = 2, fix_omega = 0, and omega = 0) with its null model (model = 2, Nsites = 2, fix_omega = 1, and omega = 1) [28]. The model assumes that the branches in the phylogeny are divided in the foreground (the one of interest for which positive selection is expected) and background (those not expected to exhibit positive selection). We set CodeML to estimate branch lengths by using random starting points (fix_blenght = −1). All models were run using the F3x4 option for expected codon frequencies based on third codon positions. Finally, the naive empirical Bayes (NEB) and the Bayes empirical bayes (BEB) approaches were used to calculate the posterior probability (PP) of each site belonging to the site class of positive selection within each alternative model.

## 3. Results

### 3.1. Phylogenetic Relationships of the Co-Deleted Genes

The clinical features of all previously analyzed patients are shown in Appendix A and described in full in Rosset et al., 2018 [18]. The deletions found in the patients are shown in Appendix A. One patient showed a typical type one deletion and the other three patients showed a typical *NF1* microdeletions.

In order to investigate the presence of the missing genes in the microdeletion region of NF1 patients in other organisms, the fourteen human genes *CRLF3, ATAD5, TEFM, ADAP2, RNF135, NF1, OMG, EVI2B, EVI2A, RAB11FIP4, COPRS, UTP6, SUZ12*, and *LRRC37B* were used as query in BLAST searches against the genomes of seven primates and another four mammalian species. The results showed that three genes were found deleted in primate species, such as *ATAD5* in *Pongo abelli, RNF135* in *Nomascus leucogenys*, and *LRRC37B* in *Callithrix Jacchus*. Figure 1 shows the deleted genes in humans and in the comparative species.

The search for deleted gene sequences in the databases NCBI and Esembl reveals that all genes are located in the same chromosomal unity in humans. The other species show some different locations, such as *Nomascus leucogenys* that has only *CRLF3* and *ADAP2* on chromosome 14 and other genes on chromosome 19, and *Mus musculus* that has all genes on chromosome 10 with the exception of *COPRS* that is located on chromosome 16. The chromosomal locations of all genes in the analyzed species are shown in Appendix A. The best models of nucleotides and protein alignments obtained by JModelTest 2.1 and ProTest 2.4 are shown in Appendix A. Phylogenetic analysis was performed for each gene of the ingroup and outgroup species. In general, trees with nucleotide and amino acid sequences showed similar topologies. However, the order of organisms was not always the same and the posterior probabilities were higher in the most phylogenies reconstructed based on nucleotide sequences (Appendix A). Only *RNF135* and *LRRC37B* genes showed better values in the trees imputed with their amino acid sequences (Appendix A). Additionally, these two genes showed considerable variation among their sequences in the species studied, (red cells from Appendix A). The results concerning *LRRC37B* trees are difficult to interpret since this gene has several pseudogenes in different species and it is not possible to infer if all these sequences are functional. *LRRC37B*, *UTP6*, and *EVI2B* trees grouped *Homo sapiens* and *Pan troglodytes*, and *Mus musculus* and *Rattus norvegicus*, however, they did not group *Canis lupus familiaris* and *Bos Taurus. SUZ12* trees did not group *Homo sapiens* and *Pan troglodytes*, instead the tree grouped *Homo sapiens* and *Gorilla gorilla gorilla*. *COPRS* trees grouped *Homo sapiens* and *Pan troglodytes* and did not group *Canis lupus familiaris* and *Bos taurus* and *Mus musculus* and *Rattus norvegicus.* Nucleotide and amino acid trees generated with *ATAD5*, *ADAP2*, *NF1*, *OMG*, *EVI2A*, and *TEFM* sequences showed all clades in correct positions. Only *ATAD5* gene was deleted in *Pongo abeli*.

*CRLF3* is probably a negative regulator of cell cycle [12] and was conserved in all analyzed species. However, the trees do not show *Homo sapiens* and *Pan troglodytes* with a common ancestor. Nucleotide sequence tree grouped *Homo sapiens* and *Gorilla gorilla gorilla*, whereas amino acid tree grouped *Homo sapiens* and *Pongo abelii*. Nucleotide tree showed better posterior values but did not group *Bos taurus* and *Canis lupus familiaris*. Finally, *RNF135* nucleotide and amino acid trees did not group *Homo sapiens* and *Pan troglodytes*; both grouped *Bos taurus* and *Canis lupus familiaris* and *Mus musculus* and *Rattus norvegicus*. This gene was not found in *Nomascus leucogenys* and was incomplete in *Tarsius syrichta*. Moreover, the *RNF135* gene could act as a transcription factor, due to the presence of zinc finger domains and the lower conservation found outside the sequence corresponding to protein domains, a common feature in transcription factors.

### 3.2. Selection Analysis of the Co-Deleted Genes

All genes presented a statistically significant value for M3 × M0 analysis, indicating ω heterogeneity among all sites (Table 1). *RNF135*, *UTP6*, *LRRC37B*, and *EVI2B* showed statistically significant values for the comparison between the M8 × M7 models (*p* < 0.005, *p* < 0.025, *p* < 0.001, and *p* < 0.025, respectively), indicating positive selection. In addition, *LRRC37B* was the only gene that also presented a statistically significant difference in the comparison of M2 × M1 models (Table 1). The naive empirical Bayes (NEB) and Bayes empirical bayes (BEB) analyses demonstrated which sites are under selection (Appendix A), however, presented statistically significant values only for sites in the *RNF135* gene. Twenty-seven amino acid sites showed positive selection, with statistical significance for the NEB model (21 sites with I-value >0.005 and 6 sites with *p*-value >0.001) (Appendix A). Figure 2 shows the positively selected sites and positions, including sites located in the RING finger domain of *RNF135*.

The *CRLF3*, *ATAD5*, *ADAP2*, *NF1*, *SUZ12*, *OMG*, *EVI2A*, *RAB11FIP4*, *TEFM*, and *CORPS* genes did not present statistically significant values for the comparisons of M2 × M1 and M8 × M7 models, indicating that they may be under purifying selection (Table 1).

## 4. Discussion

Tumor occurrence in NF1 follows the two hit mechanism proposed by Knudson in 1971 for retinoblastoma. The first hit corresponds to the germ line mutation (haploinsufficiency) that is enough to cause the initial NF1 symptoms. The second hit is somatic and stimulates tumorigenesis in specific tissues involved in NF1 disease [29]. Only a few genotype–phenotype correlations have been established to date in NF1 [30,31,32]. One of the most evident is associated with *NF1* microdeletions, independent of the second hit acquisition. The architecture of the genomic regions flanking the *NF1* gene in 17q11.2 in humans is characterized by low copy repeats (LCRs). LCRs predisposes to large deletions mediated by mutational mechanisms occurring in the germline of an unaffected parent or during mitotic postzygotic cell divisions [33]. Initially, the breakpoints of large rearrangements were characterized by microdeletion junction-specific PCR assay of 54 NF1 patients. The recombinant event occurred in 46% of cases in a recombination hotspot region of the flanking NF1REPs, also causing three intervals of recombinant events [34]. Later, four types of microdeletions (type 1, 2, 3, and atypical) were identified; the most frequent type of microdeletion is type 1 *NF1* deletion which encompass 1.4 Mb and include 14 protein-coding genes as well as four microRNA genes (Appendix A) [35,36,37]. The non-allelic homologous recombination (NAHR) events causing type 1 *NF1* deletions are mediated by the low copy repeats, NF1-REPa, and NF1-REPc [36]. Type 2 deletions encompass only 1.2 Mb and are associated with hemizygosity for 13 protein-coding genes. They are also mediated by NAHR but their breakpoints are located within *SUZ12* and its highly homologous pseudogene *SUZ12P1* which flank NF1-REPc and NF1-REPa, respectively (Appendix A) [37]. Type 3 *NF1* deletions are very rare; these 1.0 Mb deletions occur in only 1–4% of all patients with gross *NF1* deletions and are mediated by NAHR between NF1-REPb and NF1-REPc leading to hemizygosity for a total of nine protein-coding genes (Appendix A). Atypical large *NF1* deletions do not exhibit recurrent breakpoints and are variable considering the number of genes located within the deleted region [38].

In our previously studied cohort [18], one of the NF1 probands showed the well described *NF1* type 1 microdeletion. Although the other three patients showed *NF1* deletion breakpoints similar to type 2 and type 3 microdeletions, they were considered atypical (Appendix A). One gene in the region between NF1-REPa and NF1-REPb, which is variably included in microdeletions and is deleted in four of our patients, is ring finger protein 135 (*RNF135*). *RNF135* codifies a protein that contains a RING finger domain at the N terminus and a B30.2/SPRY domain at the C terminus. RING domains are specialized zinc-finger motifs that can have ubiquitin and sumo ligase activity [39]. *RNF135* loss-of-function mutations, as well as an NF1-REPa to NF1-REPb deletion including this gene, have been implicated in an overgrowth syndrome which includes tall stature, macrocephaly, dysmorphic features, and variable additional features, including learning disability [39]. The dysmorphic facial character as seen in patients with *NF1* microdeletions is generally missing in patients with intragenic *NF1* mutations. One of the patients from our previous study (Appendix A) had an *NF1* deletion including NF1-REPa to NF1-REPb and *RNF135* and had several dysmorphic features as well as tall stature. However, since only the study conducted by Douglas et al. (2007) [39] analyzed patients with intragenic *RNF135* mutations but lacking *NF1* microdeletions, more studies are necessary to assess the role of *RNF135* co-deletion in NF1. In our study, a different approach was applied to try to elucidate the importance of this gene in *NF1* microdeletions. The phylogenetic analysis revealed the evolutionary relationships of each gene among the primate species analyzed. These results showed that some genes are not in agreement with the species tree, suggesting distinct evolutionary constraint between them, which could be expected for genes that present divergence of function during species evolution and are under distinct selection pressure. The distinct statistical values for clades on the phylogenies considering nucleotide and protein sequences could be explained by the differences in nucleotide and protein variation. These differences could be explained by the high synonymous substitution found for some genes. The phylogenetic trees were used to estimate the dN/dS value for molecular evolutionary rate analysis and revealed a distinct selection pattern for each gene. Most genes are under purifying or negative selection, but some present some positively selected sites.

Our results indicate that there is heterogeneity of ω between the sites of all analyzed genes in the different species. This indicates that, in the same position, the amino acids vary among the different groups investigated. Four genes present evidence of being under positive selection (*RNF135*, *UTP6*, *LRRC37B*, and *EVI2B*), which possibly increased their adaptive values. Of these, the only gene that presented statistically significant differences for its sites in the NEB model was *RNF135* (Appendix A). The protein RNF135 showed twenty-seven sites under positive selection, and three of them are located in the RING finger domain of the protein. The second positively selected residue is a glutamic acid, with a negative net charge in humans and other primates (*Macaca mulatta*, *Callithrix Jacchus, Pongo abelii*, and *Pan troglodytes*). In organisms that are phylogenetically more distant to humans (e.g., *Mus musculus*), this site has a positively charged amino acid. The third residue is an aspartic acid, a negative amino acid in the human protein. In the primates *Pan troglodytes*, *Pongo abelii*, and *Macaca mulatta*, the neutral amino acid glicine was selected; in *Mus musculus* the amino acid in this position is an arginine. Finally, the fourth selected residue is an arginine in humans and other main primates; in *Mus musculus*, a valine, a polar hidrofobic amino acid, is in this position. These differences in amino acids and their charges could have been selected in humans and other primates to allow better tridimensional conformation of the RNF135 protein and, consequently, a higher efficiency in its ubiquitination function. Most of the ring finger proteins contain two zinc atoms grouped with cistein or cistein-histidin rich clusters. The consensus sequence of this domain is C – X2 – C – X9–39 – C – X1–3 – C – X2 – C – X4–48 – C – X2 – C [40]. The *RNF135* gene has this consensus region, represented in Figure 2, which encompass the second, third, and fourth sites under positive selection (shown in red in Figure 2). Genes under this type of selection have important functions and these data support that *RNF135* haploinsufficiency contributes to human disease. Other genes showed positive selection in our study (*UTP6, LRRC37B*, and *EVI2B*), but without a statistically significant difference in the NEB model (Appendix A).

The other ten genes showed evidence of being under purifying selection, which is related to the importance of maintaining their function during the evolution of organisms. They are fairly conserved genes that are present in all organisms investigated, with the exception of the *ATAD5* gene in *Pongo abelli*. These genes in the co-deleted region may also contribute to the phenotype in microdeleted patients. *SUZ12* (also known as *JJAZ1*) is critical in embryonic development [41]. There are evidences that biallelic *SUZ12* loss promotes malignant peripheral nerve sheath tumor (MPNST) progression in NF1 [42]. The SUZ12 protein is a component of the Polycomb repressive complex 2 (PRC2) and is involved in the epigenetic silencing of many different genes by establishing di- and tri-methylation of histone H3 lysine 27 [43]. Loss of histone H3 lysine 27 trimethylation was observed in 50–70% of MPNSTs. By contrast, in benign neurofibromas, the H3 lysine 27 trimethylation is retained, serving as a diagnostic marker for malignant transformation [44]. Another gene, *OMG*, which encodes the oligodendrocyte myelin glycoprotein is an important inhibitor of neurite overgrowth [45]. Haploinsufficiency of the *OMG* gene has been proposed to be associated with learning disability. Cardiovascular malformations observed in patients with *NF1* microdeletions could be related to hemizygosity of the *ADAP2* gene. This conclusion is drawn from the observation that *ADAP2* is highly expressed during early stages of heart development in both mice and humans [46]. Haploinsufficiency of *UTP6* seems to reduce cellular apoptosis, increasing the risk of tumor development [47]. *ATAD5* (ATPase family AAA domain-containing protein 5) is involved in the stabilization of stalled DNA replication forks by regulating proliferating cell nuclear antigen (PCNA) ubiquitination during DNA damage bypass, thereby promoting the exchange of a low-fidelity translesion polymerase back to a high-fidelity replication polymerase [48]. Embryonic fibroblasts from more than 90% of haploinsufficient Atad5+/m mice developed tumors such as sarcomas, carcinomas, and adenocarcinomas that exhibited high levels of genomic instability [49].

## 5. Conclusions

In our study, we found that four of the frequently co-deleted genes in NF1 patients with microdeletions present evidence of being under positive selection, including specific amino acid sites of RNF135 protein in its main functional domain (RING finger), reinforcing its importance and contribution to human disease. Moreover, the other ten co-deleted genes showed evidence of being under purifying selection, which is related to the importance of maintaining their function during the evolution of organisms. Our in silico analysis could help clarify the function and contribution of the co-deleted genes to disease and, in the future, co-deleted genes could possibly be included in the *NF1* molecular diagnostic workup.

## Figures and Tables

**Figure 1 genes-10-00839-f001:**
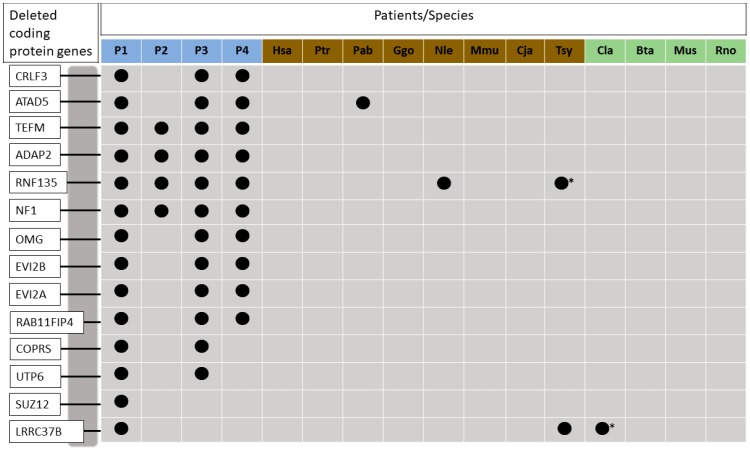
Deleted genes analyzed in this study and in comparative species used for the phylogenetic analysis. Blue squares represent patients with *NF1* microdeletions: patient 1, patient 2, patient 3, and patient 4, respectively. Brown squares represent ingroup species: *Homo sapiens, Pan troglodytes, Pongo abelii, Gorilla gorila gorila, Nomascus leucogenys, Macaca mulatta, Callithri Jacchus e Tarsius syrichata*, respectively. Green squares represent outgroup species: *Canis lúpus familiaris, Bos taurus, Mus musculus* e *Rattus norvegicus*, respectively. Deleted genes in each species are represented with dark circles. Asterisk indicates that the *Tarsius syrichata RNF135* gene, and *Canis lupus LRRC37B* are absent because their sequences are mostly incomplete.

**Figure 2 genes-10-00839-f002:**
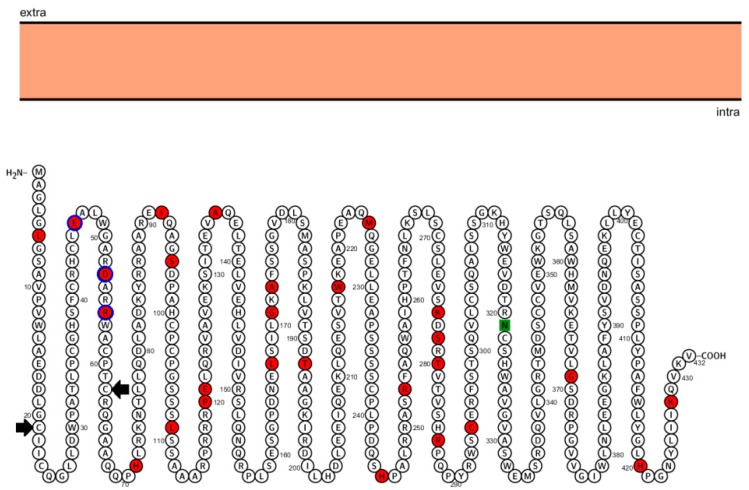
Sequence of human protein RNF135 and their positive selected sites. Representation of the structure of the cytoplasmic protein RNF135 and its 432 amino acids. The orange rectangle represents the cell membrane. The small circles represent the amino acids of the protein, with the names indicated by the letters contained within the circles. The beginning and end of the sequence corresponding to the RING finger domain are indicated by the first and second black arrows respectively. The circles painted in red represent the 27 amino acids that are under positive selection. The green square represents the *N*-glycosilated motif.

**Table 1 genes-10-00839-t001:** CodeML analysis of the entire dataset using the site models.

Gene Family	Model/Likelihood	Comparison	Parameters/Significance
*CRLF3*	M0/-3789,882728	M3 vs. M0	2ΔL = 31.37 (df = 4)/*p* << 0.001 *
	M1/-3774,195335	M2 vs. M1	2ΔL = −6.51(df = 2)/*p* < 0.99
	M2/-3777,450354	M8 vs. M7	2ΔL = 0.79(df = 2)/*p* < 0.5
	M3/-3774,195335		
	M7/-3774,904179		
	M8/-3774,50421		
*ATAD5*	M0/-22006,87329	M3 vs. M0	2ΔL = 301.11 (df = 4)/*p* < 0.001 *
	M1/-21865,9053	M2 vs. M1	2ΔL = 0 (df = 2)/*p* < 0.99
	M2/-21865,9053	M8 vs. M7	2ΔL = 4.26 (df = 2)/*p* < 0.5
	M3/-21856,31708		
	M7/-21859,57933		
	M8/-21857,4487		
*ADAP2*	M0/-4320,93576	M3 vs. M0	2ΔL = 83.94 (df = 4)/*p* < 0.001 *
	M1/-4283,476747	M2 vs. M1	2ΔL = 0 (df = 2)/*p* < 0.99
	M2/-4283,476747	M8 vs. M7	2ΔL = 0.176 (df = 2)/*p* < 0.99
	M3/-4278,962315		
	M7/-4279,346509		
	M8/-4279,258106		
*RNF135*	M0/-5802,616454	M3 vs. M0	2ΔL = 116.5 (df = 4)/*p* < 0.001 *
	M1/-5745,76494	M2 vs. M1	2ΔL = 2.80 (df = 2)/*p* < 0.1
	M2/-5744,363479	M8 vs. M7	2ΔL = 11.65 (df = 2)/*p* < 0.005 *
	M3/-5744,351541		
	M7/-5750,260024		
	M8/-5744,430178		
*NF1*	M0/-20275,12645	M3 vs. M0	2ΔL = 73.08 (df = 4)/*p* < 0.001 *
	M1/-20240,46408	M2 vs. M1	2ΔL = −4 × 10^−6^ (df = 2)/*p* < 0.99
	M2/-20240,46408	M8 vs. M7	2ΔL = 1.57 (df = 2)/*p* < 0.99
	M3/-20238,58367	M3 vs. M0	
	M7/-20239,50865		
	M8/-20238,71939		
*UTP6*	M0/-6507,150936	M3 vs. M0	2ΔL = 154.60 (df = 4)/*p* < 0.001 *
	M1/-6433,008949	M2 vs. M1	2ΔL = 0 (df = 2)/*p* < 0.99
	M2/-6433,008949	M8 vs. M7	2ΔL = 8.52 (df = 2)/*p* < 0.025 *
	M3/-6429,850049		
	M7/-6434,194703		
	M8/-6429,931052		
*SUZ12*	M0/-4810,097946	M3 vs. M0	2ΔL = 14.64 (df = 4)/*p* < 0.005 *
	M1/-4803,610839	M2 vs. M1	2ΔL = 0 (df = 2)/*p* < 0.99
	M2/-4803,610839	M8 vs. M7	2ΔL = −0.000288 (df = 2)/*p* < 0.99
	M3/-4802,77363		
	M7/-4802,776934		
	M8/-4802,777078		
*OMG*	M0/-3421,827547	M3 vs. M0	2ΔL = 20.99 (df = 4)/*p* < 0.001 *
	M1/-3412,432711	M2 vs. M1	2ΔL = 0 (df = 2)/*p* < 0.99
	M2/-3393,13422	M8 vs. M7	2ΔL = 0.018 (df = 2)/*p* < 0.99
	M3/-3412,246309		
	M7/-3412,265355		
	M8/-3412,256186		
*LRRC37B*	M0/-16071,4708	M3 vs. M0	2ΔL = 120.7 (df = 4)/*p* < 0.001 *
	M1/-16018,91532	M2 vs. M1	2ΔL = 9.87 (df = 2)/*p* < 0.005 *
	M2/-16013,97878	M8 vs. M7	2ΔL = 13.30 (df = 2)/*p* < 0.001 *
	M3/-16011,07676		
	M7/-16018,02362		
	M8/-16011,36949		
*EVI2A*	M0/-2792,694611	M3 vs. M0	2ΔL = 44.45 (df = 4)/*p* < 0.001 *
	M1/-2771,253237	M2 vs. M1	2ΔL = 1.32 (df = 2)/*p* < 0.99
	M2/-2770,590286	M8 vs. M7	2ΔL = 4.28 (df = 2)/*p* < 0.1
	M3/-2770,464944		
	M7/-2772,654596		
	M8/-2770,510367		
*EVI2B*	M0/-5998,097719	M3 vs. M0	2ΔL = 71.86 (df = 4)/*p* < 0.001 *
	M1/-5964,792227	M2 vs. M1	2ΔL = 1.85 (df = 2)/*p* < 0.1
	M2/-5963,862998	M8 vs. M7	2ΔL = 6.25 (df = 2)/*p* < 0.025 *
	M3/-5962,167576		
	M7/-5965,886294		
	M8/-5962,759212		
*RAB11FIP4*	M0/-8519,576811	M3 vs. M0	2ΔL = 214.76 (df = 4)/*p* < 0.001 *
	M1/-8500,408316	M2 vs. M1	2ΔL = 0 (df = 2)/*p* < 0.99
	M2/-8500,408316	M8 vs. M7	2ΔL = 3.27 (df = 2)/*p* < 0.1
	M3/-8412,192643		
	M7/-8412,081211		
	M8/-8410,442254		
*TEFM*	M0/-4523,914053	M3 vs. M0	2ΔL = 95.54 (df = 4)/*p* < 0.001 *
	M1/-4479,041925	M2 vs. M1	2ΔL = 2.70 (df = 2)/*p* < 0.1
	M2/-4477,690271	M8 vs. M7	2ΔL = 3.91 (df = 2)/*p* < 0.1
	M3/-4476,141837		
	M7/-4478,690601		
	M8/-4476,734719		
*CORPS*	M0/-2021,525059	M3 vs. M0	2ΔL = 39.62 (df = 4)/*p* < 0.001 *
	M1/-2001,718453	M2 vs. M1	2ΔL = 0 (df = 2/*p* < 0.99
	M2/-2001,718453	M8 vs. M7	2ΔL = 0.26 (df = 2)/*p* < 0.5
	M3/-2001,71364		
	M7/-2001,856893		
	M8/-2001,724449		

* Statistical significant result obtained by Fisher chi-square.

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
