# Peer review of "The Role of Co-Deleted Genes in Neurofibromatosis Type 1 Microdeletions: An Evolutive Approach"

_genes, 2019, doi:10.3390/genes10110839_

Round 1
Reviewer 1 Report
The article presents an interesting approach to the study and analysis of NF1 deletions. The evolutive approach is well described and the methods are sound. Results and discussion are well articulated.
On the present article it might be relevant to cite some of the initial papers describing NF1 deletion breakpoint mapping papers, one of these is:
Hum Mol Genet. 2001 Jun 15;10(13):1387-92.
Recombination hotspot in NF1 microdeletion patients.
López-Correa C1, Dorschner M, Brems H, Lázaro C, Clementi M, Upadhyaya M, Dooijes D, Moog U, Kehrer-Sawatzki H, Rutkowski JL, Fryns JP, Marynen P, Stephens K, Legius E.
Also, as part of the paper it will be useful to add more clinical information about the 4 NF1 deletion patients included in the study. Are their phenotype and clinical presentations similar?
Aside from these minor edits there are no major changes that should be done to the paper. It presents an innovative approach to use available primate sequences to analyze the content and potential role that genes contained in NF1 deletions have on the more severe phenotype of these patients
Author Response
On the present article it might be relevant to cite some of the initial papers describing NF1 deletion breakpoint mapping papers, one of these is:
Hum Mol Genet. 2001 Jun 15;10(13):1387-92.
Recombination hotspot in NF1 microdeletion patients.
López-Correa C1, Dorschner M, Brems H, Lázaro C, Clementi M, Upadhyaya M, Dooijes D, Moog U, Kehrer-Sawatzki H, Rutkowski JL, Fryns JP, Marynen P, Stephens K, Legius E.
Answer: We thank the reviewer and agree with this suggestion. The paper was cited in the discussion section, page 8, lines 238 – 241.
Also, as part of the paper, it will be useful to add more clinical information about the 4 NF1 deletion patients included in the study. Are their phenotype and clinical presentations similar?
Answer: We thank the reviewer and emphasize that the Supplementary Table 1 describes the clinical presentation and type of mutation of the 4 patients. Additionally, Supplementary Table 1 includes a comparison of symptoms and their initial age along with other information.
Reviewer 2 Report
The paper mentions phenotypes of microdeletion patients, such as stature/overgrowth, learning difficulties, malignacies etc. It would improve the paper if there was an additional table of these features for each of the 4 patients studied, along with their age. This would allow comparisons to be made with the features present compared to the genes deleted. A photograph of the dysmorphic features would be interesting, provided there was appropriate consent.
Author Response
The paper mentions phenotypes of microdeletion patients, such as stature/overgrowth, learning difficulties, malignancies etc. It would improve the paper if there was an additional table of these features for each of the 4 patients studied, along with their age. This would allow comparisons to be made with the features present compared to the genes deleted.
Answer: We thank the reviewer and emphasize that the Supplementary Table 1 describes the clinical presentation and type of mutation of the 4 patients. Additionally, this Supplementary Table 1 includes a comparison of symptoms and their initial age along with other information. We did not make comparisons of the features according to deleted genes because there is only one patient with each type of microdeletion. To make reliable comparisons a larger sample would be necessary.
A photograph of the dysmorphic features would be interesting, provided there was appropriate consent.
Answer: We thank the reviewer and agree that photographs would be interesting to illustrate the phenotypic characteristics of patients. Unfortunately, we do not have the appropriate consent of patients to expose their photographs. The patients recruited to this study live in different cities, distant to the reference center Hospital de Clínicas de Porto Alegre (HCPA), and they have limited conditions to come to HCPA. Therefore, we are not able to recruit these patients again in five days.
Reviewer 3 Report
Reis et al present an important study on NF1 due to microdeletions. Their study is well written and adds to the current knowledge of phenotypic and genetic heterogeneity in this condition. The authors are encouraged to expand on the clinical manifestations of these probands, namely tanner staging, absence of thelarche if any, evidence of growth hormone excess etc.
Author Response
The authors are encouraged to expand on the clinical manifestations of these probands, namely tanner staging, absence of thelarche if any, evidence of growth hormone excess etc.
Answer: We thank the reviewer. Unfortunately, the specific symptoms of thelarche and growth hormone dosage were not evaluated. To our knowledge, there is no evidence that the microdeletions are responsible for these symptoms. In general, the NF1 patients develop most of the symptoms during puberty, but this does not seem to be related to type of mutations.
Reviewer 4 Report
The authors present an interesting approach to explain the phenotypoic variability in Neurofibromatosis type I by confining the analysis to patients with large deletions and examining the conservation of the co-deleted genes in the region.
The article is well written and very detailed in the methods and results achieved, but contains a main flaw, which is the attempt to explain the NF1 phenotypes with the haploinsufficiency alone. NF1 is a well known oncosuppressor gene, which has been demonstrated to be inactivated also in target tissues by a second somatic hit. If the oncosuppressor mechanism has not been demonstrated for the main phenotypes related to the large deletions (intellectual disability, for example), and probably does not account for that phenotype, it should be mentioned that it is crucial for most of the other features, like neurofibromas or cancer in NF1. A paragraph in the introduction and in the discussion sections would help the reader to understand that haploinsufficiency is not the only (and possibly not the main) phenotypic mechanism in neurofibromatosis type I and not the only mutational event.
Shortening of the discussion would help the reader to focus in the main messages of the authors
Some mistakes in the English language: past tenses, among vs between, singular vs plural
Author Response
NF1 is a well known oncosuppressor gene, which has been demonstrated to be inactivated also in target tissues by a second somatic hit. If the oncosuppressor mechanism has not been demonstrated for the main phenotypes related to the large deletions (intellectual disability, for example), and probably does not account for that phenotype, it should be mentioned that it is crucial for most of the other features, like neurofibromas or cancer in NF1. A paragraph in the introduction and in the discussion sections would help the reader to understand that haploinsufficiency is not the only (and possibly not the main) phenotypic mechanism in neurofibromatosis type I and not the only mutational event.
Answer: We included some sentences in the introduction (lines 35-36) and discussion (lines 230-235) sections about haploinsufficiency, second somatic hit mutations and how they act on the symptomatology of NF1.
Shortening of the discussion would help the reader to focus in the main messages of the authors
Answer: We thank the reviewer and shortened the discussion section as requested.
Some mistakes in the English language: past tenses, among vs between, singular vs plural
Answer: The English language of the manuscript was has been reviewed, and the mistakes were corrected.